# Genome-Wide Prediction and Analysis of *Oryza* Species *NRP* Genes in Rice Blast Resistance

**DOI:** 10.3390/ijms231911967

**Published:** 2022-10-08

**Authors:** Dong Liang, Junjie Yu, Tianqiao Song, Rongsheng Zhang, Yan Du, Mina Yu, Huijuan Cao, Xiayan Pan, Junqing Qiao, Youzhou Liu, Zhongqiang Qi, Yongfeng Liu

**Affiliations:** Institute of Plant Protection, Jiangsu Academy of Agricultural Sciences (JAAS), Nanjing 210014, China

**Keywords:** *Oryza* species, *NRP* genes, conserved evolution, bZIP50 TF, rice immune response

## Abstract

Members of the N-rich proteins (NRPs) gene family play important roles in the plant endoplasmic reticulum stress in response, which can be triggered by plant pathogens’ infection. Previous studies of the *NRP* gene family have been limited to only a few plants, such as soybean and *Arabidopsis thaliana*. Thus, their evolutionary characteristics in the *Oryza* species and biological functions in rice defense against the pathogenic fungus *Magnaporthe oryzae* have remained unexplored. In the present study, we demonstrated that the *NRP* genes family may have originated in the early stages of plant evolution, and that they have been strongly conserved during the evolution of the *Oryza* species. Domain organization of NRPs was found to be highly conserved within but not between subgroups. *OsNRP1*, an *NRP* gene in the *Oryza sativa japonica* group, was specifically up-regulated during the early stages of rice-*M. oryzae* interactions-inhibited *M. oryzae* infection. Predicted protein-protein interaction networks and transcription-factor binding sites revealed a candidate interactor, bZIP50, which may be involved in *OsNRP1-*mediated rice resistance against *M. oryzae* infection. Taken together, our results established a basis for future studies of the *NRP* gene family and provided molecular insights into rice immune responses to *M. oryzae*.

## 1. Introduction

Rice is a crucial crop that is responsible for feeding more than half of the world’s population [1]. However, a variety of extreme environmental conditions, (such as drought, salinity and extreme temperatures) negatively affect rice plant growth. Additionally, rice blast, which is caused by the hemi-biotrophic fungal pathogen *Magnaporthe oryzae*, can reduce rice yield by 30% [2].

To inhibit pathogens’ invasion, host plants have evolved a two-layer immune system [3,4]. The first is determined by plant pattern recognition receptors (PRRs). Through recognizing pathogen-associated molecular patterns (PAMPs) of pathogens (such as PGN and chitin oligosaccharide), plant PRRs activate PAMP-triggered immunity (PTI) for the inhibition of pathogens’ colonization. Pathogens can correspondingly release various effector proteins into host cells to evade or subvert host PTI defenses [5]. The second layer plant immune system is governed by resistance (R) proteins, which mainly have nucleotide-binding site leucine-rich repeat (NBS-LRR) domain architecture. Through directly or indirectly binding to avirulence (Avr) effectors of pathogens, R proteins are induced and activate a rapid response, also known as effector-triggered immunity (ETI) [6].

Rice resistance based on NBS-LRR proteins is often overcome within five years due to the rapidly evolving and highly variable effectors of blast fungus [7]. In contrast, multiple defense-regulator (DR) genes also confer partial but broad and durable resistance. Until now, dozens of DR genes were found that maintain this durable resistance by guaranteeing effective signal transduction and being responsible for downstream immune responses [8,9]. These downstream immune responses include accumulation of antimicrobial compounds or hormones, reactive oxygen species (ROS) bursts, and activation of programmed cell death (PCD) [10,11]. The endoplasmic reticulum (ER) is a key organelle involved in the activation of the stress response [12]. During pathogen-plant interactions, effector proteins target not only host defense-related proteins, but also ER resident proteins, causing ER-stress [13,14]. This triggers the unfolded protein response (UPR) to inhibit the accumulation of unfolded/misfolded proteins. If levels of excessive unfold/misfold surpass a threshold level, the PCD pathway is activated [15].

ER stress and osmotic stress signaling, which form an integrated major response to ER stress-activated cell death, converges on N-rich proteins (NRPs). This integrated signaling pathway was originally identified in soybean (*Glycine max*, *L*.) during the early stage of the hypersensitive response, which was activated by infection with avirulent bacteria [14]. NRP proteins were characterized by their NRP domain, namely development and cell death (DCD) domain, which is plant-specific and composed of around 130 amino acids. However, NRP proteins appear to be absent in bacteria and fungi. This supports the hypothesis that NRP proteins are plant-specific and were present at the start of plant evolutionary history, prior to the separation of higher plants. The NRP protein family can be divided into four subgroups based on the location of the NRP domain. Ref. [16] detected that amino acids of soybean NRP proteins formed alpha-helix and remaining amino acids were used to generate beta strands. Meanwhile, the N-terminal halves of NRP proteins are asparagine-rich (24%), whereas the C-terminus regions contain the NRP domain [16,17]. These proteins include an FGLP and an LFL motif at the N-terminus and a PAQV and a PLxE motif toward the C-terminus.

Not only environmental changes but also pathogen infection can disturb ER homeostasis in the host plant [18]. Once host ER homeostasis is disrupted, two type ER-stress sensors were activated to reduce the content of unfolded or misfolded proteins so that the host ER homeostasis is restored. Protein kinase RNA-like ER kinase (PERK), inositol-requiring protein-1 (IRE1), and activating transcription factor-6 (ATF6) were assigned to type 1 ER-stress sensors; Type 2 includes bZIP transcription factor (TF) bZIP60 [19].

However, if ER stress is sustained, the NRP-mediated PCD signaling pathway is activated. For example, in soybean, the ER stress- and osmotic stress-induced TF GmERD15 regulates expression of *NRP* genes through binding to NRP promoters [20]. NRP proteins transduce hallmark PCD signals to activate the PCD signaling cascade, which induces expression of the TFs *GmNAC81* and *GmNAC30*. GmNAC81 and GmNAC30 transactivate vacuolar processing enzyme (VPE) by heterodimerization, then induce plant-specific PCD [21].

Thanks to the rapid development of high-throughput sequencing method in the last decade, high-quality and complete genome assemblies of many plant species were obtained, which help researchers to identify gene families that might link to plant development, abiotic or biotic stress response, and thus have a comprehensive understanding of them. For example, [22] identified 33 encoding genes of β-Ketoacyl CoA synthetase (KCS) through BLAST and hidden Markov model search against barley genome and found some of them may determine the synthesis of very-long-chain fatty acids (VLCFAs), which affect the formation of epidermal wax under drought stress. Moreover, similar methods were also applied for detection of proline-rich extensin-like receptor kinases (PERKs) gene family in wheat, which revealed 37 PERK genes and eight of them were up-regulated in response to drought and heat stress [23]. Beyond identification of encoding gene families, globally predicted natural antisense transcripts (NATs) and long non-coding RNAs (lncRNAs) were predicted through mapping RNA-seq reads on corresponding reference genome assemblies [24,25], which revealed regulatory mechanisms of compounds biosynthesis (anthocyanin and phenylpropanoid) in *Salvia miltiorrhiza* and abiotic stress response in *Capsicum annuum*.

The functions of NRP proteins in soybean and *Arabidopsis thaliana* in response to diverse stressors have previously received a great deal of attention. However, few studies have reported the functions of NRP proteins in rice resistance to blast fungus. The roles and biological functions of NRP proteins in other rice defense responses also remain unclear. The present study aimed to shed light on the evolutionary history, putative interaction partners, expression patterns, and biological functions of NRPs in rice resistance. The results revealed conserved evolution of the NRP protein family and synergistic differentiation between *Oryza* species. We also found that *OsNRP1*, an *NRP* gene in the *Oryza sativa japonica* group, was specifically up-regulated at the early infection stage and enhanced rice resistance to *M. oryzae* attack. Furthermore, OsNRP1 was also predicted to interact with IRE1, bZIP TFs and NAC-domain containing proteins. Our findings provide a basis for further investigation of NRP protein functions and regulatory mechanisms in rice resistance.

## 2. Results

### 2.1. Whole-Genome Characterization and Phylogenic Analysis of NRP Genes in Oryza Species

Two strategies were independently applied for the identification of *NRP* genes in *Oryza* species. First, a hidden Markov model approach (HMM) was used to search against 12 *Oryza* species and soybean proteomics, which were retrieved from the Ensembl plant database (https://plants.ensembl.org/, (accessed on 5 July 2022)), based on the NRP domain HMM model file (Pfam accession: PF10539). Second, 26 characterized plant NRP genes were used as queries in BLASTP searches against the 13 proteomes using an e-value cutoff of 1 × 10^−5^. To further verify the reliability of candidates *NRP* genes, the Conserved Domains Database (CDD) tool was used to validated the completeness of the NRP domain in each candidate. A total of 136 complete, non-redundant *NRP* genes were detected in 12 *Oryza* species and soybean (Table 1), with each *Oryza* species containing between nine and 13 homologs, and 15 in soybean (Appendix A and Figure 1a). This demonstrated wide and conserved distribution of *NRP* genes among *Oryza* species.

We investigate the evolutionary history of *NRP* genes in *Oryza* species by constructing a maximum likelihood (ML) phylogenetic tree (Figure 1b). A total of 136 putative *NRP* genes in the *Oryza* species and soybean were identified. These genes were categorized into four subgroups (I-IV). *NRP* genes from each *Oryza* species and soybean were present in all four groups, suggesting that four ancestral *NRP* subgroups divided in the most recent common ancestor (MRCA) of *Oryza* species and soybean. No clustered pairs of *NRP* genes were detected in any *Oryza* species, indicating a lack of species-specific duplication events. In contrast, paralogous pairs of *G. max NRP* genes in each subgroup revealed that this family underwent several duplication events in soybean. After scanning the domain architecture of each protein, we found tandem Kelch motifs located in C-terminus of all subgroup II members except *OINRP3*, *GmNRP1*, and *GmNRP11*. The preference of C-terminus location was found among *NRP* genes in subgroup I, which differed from other *NRP* genes. Notably, NRP domain duplications were also detected in NRP proteins in subgroup III and IV.

### 2.2. Synteny Analysis of the NRP Gene Family in Oryza Species

There were 12, 13, 11, 11, and 15 *NRP* genes in the *O. sativa japonica* group, the *O. sativa indica* group, *Oryza punctata*, *Oryza barthii*, and *Glycine max*, respectively. To further investigate the orthologous relationships, a synteny analysis was conducted for *NRP* genes in these species (Figure 2a and Appendix A). The *O. sativa japonica* group genome shared 10, 11, and 11 syntenic *NRP* genes with the *O. sativa indica* group, *O. barthii*, and *O. punctata*, respectively. The *O. sativa indica* group genome had 11 and 10 syntenic genes with *O. barthii* and *O. punctata*, respectively, and there were 11 syntenic *NRP* genes between *O. barthii* and *O. punctata*. These results indicated a high level of conservation of *NRP* gene family after differentiation of *Oryza* species. In contrast, only *GmNRP4* and *GmNRP12* showed a syntenic relationship with corresponding genes in *Oryza* species, suggesting more extensive divergence between NRP genes in soybean and *Oryza* species. An intraspecific synteny analysis of the *O. sativa japonica* group revealed that only *OsNRP3* and *OsNRP8* had a syntenic relationship (Figure 2b), indicating that there was no species-specific expansion of the *NRP* gene family. These results together with the phylogenic analysis (Figure 1b) indicated that the putative duplication occurred in the MRCA of *Oryza* species.

The ratio of nonsynonymous (Ka) to synonymous (Ks) nucleotide substitution rates is a common measurement used to distinguish between selective pressures on protein-coding genes and to assess their evolutionary rates [26]. When Ka/Ks ≈ 0, a gene is considered to be under neutral selection, whereas Ka/Ks < 1 indicates purifying selection and if Ka/Ks > 1 indicates positive selection [27]. Ka/Ks value of the syntenic *NRP* gene pairs detected above were nearly all <1 (Appendix A), suggesting that the *NRP* gene families in these species had evolved under purifying selection. However, the syntenic gene pair *OsNRP4*/*OsinNRP6* (which was also syntenic with *OsNRP2*) had a Ka/Ks ratio of 1.79, revealing that *OsNRP4* may have been under positive selection and may have a novel function.

### 2.3. NRP Genes in the O. sativa japonica Group were Differentially Expressed during M. oryzae Infection

To analyze the expression patterns of *O. sativa japonica* group *NRP* genes (*OsNRPs*) during *M. oryzae* infection, we calculated expression values using the data from our previously RNA sequencing (RNA-seq) [28]. In that study, *O. sativa* L. ssp. *japonica* cv. ‘Nipponbare’ (Nip) plants were infected with three *M. oryzae* strains (A248, B235, and C162). Seven *OsNRP* genes were found to be significantly up-regulated during the interactions between Nip and *M. oryzae*. As Figure 3a showed, *OsNRP2*, *OsNRP3*, *OsNRP4*, *OsNRP5* and *OsNRP8* were significantly up-regulated at 24 h post infection (hpi). In addition, *OsNRP1* and *OsNRP6* were significantly up-regulated at 8 hpi with *M. oryzae* A248 or B235. Due to our rice the genetic transformation system relied on wild-type cultivar ‘Zhonghua 11’ (ZH11), which was used as the background cultivar. We performed quantitative reverse transcription (qRT)-PCR assays to validate the expression patterns of seven *OsNRPs* candidates during interactions between ZH11 and Guy11 (Figure 3b). *OsNRP2*, *OsNRP3*, and *OsNRP4* were down-regulated during infection. *OsNRP6* and *OsNRP8* were up-regulated at 8, 48, 72, and 96 hpi, whereas *OsNRP6* was up-regulated at 24, 72, and 96 hpi. Notably, *OsNRP1* (*LOC_Os01g36950*) was specifically up-regulated at eight hpi but was barely detected at other timepoints. These results suggested that *OsNRP1* may play a role in the plant defense response at 8 hpi.

### 2.4. OsNRP1, a NRP Protein in the O. sativa japonica Group, may Enhance Blast Fungus Resistance

The high expression levels of *OsNRP1* during the early infection stage suggested that *OsNRP1* may be involved in blast resistance. To validate this hypothesis, a transgenic *OsNRP1*-overexpression line (*OsNRP1^OX^*) was constructed. We obtained a total of 22 independent transgenic T_1_ lines. Through qRT-PCR validation of each T_1_ lines, we found that *OsNRP1* was expressed at significantly level (around four- to 12-fold higher than in the wild-type) in three T_1_ lines: PXQ8-4, PXQ8-5, and PXQ8-17 (Figure 4d). These three lines were selected for inoculation assays by *M. oryzae* Guy11. Seven-days after in vitro inoculation, the total lesion length on ZH11 was 2.7 cm (1.5 cm + 1.2 cm), which was significantly longer than the total lesion lengths on the *OsNRP1^OX^* plant (1.5, 1.4, and 1.7 cm on PXQ8-4, PXQ8-5, and PXQ8-17, respectively) (Figure 4a). Similar results were observed after in vivo inoculation; the diseased leaf area was significantly smaller on PXQ8-4, PXQ8-5 and PXQ8-17 than on ZH11 (Figure 4b,c). These results suggested that *OsNRP1* might be activated by blast pathogen infection and could enhance rice resistance against *M. oryzae*.

### 2.5. Predicted Protein-Protein Interaction (PPI) Network of OsNRP1

NRPs require interacting partners to activate the NRP-mediated cell-death signaling pathway [20]. To further analyze the mechanisms of *OsNRP1*-mediated resistance to rice blast, a predicted interaction network was constructed for OsNRP1 using STRING website (Figure 5a). We identified 10 candidate proteins that may interact with OsNRP1 (Appendix A), including a bZIP TF (bZIP50), Serine/threonine-protein kinase (IRE1), Zinc finger domain-containing stress-associated proteins (SAP16, SAP3 and SAP5), a NAC domain-containing protein (Q5Z7Q4), and cold shock domain-containing protein (CSP1). We also investigated the PPI network of an OsNRP1 ortholog in *Arabidopsis thaliana*, AT5G42050 (Figure 5b). Notably, a bZIP TF (bZIP60) and NAC domain-containing proteins (NAC062 and NAC036) were also detected in that network.

Gene Ontology (GO) annotation enrichment analysis was conducted for the putative OsNRP1 interaction partners. This analysis revealed that ‘Response to stress’, ‘Response to organic substance’, and ‘Endoplasmic reticulum unfolded protein response’ were the major enriched terms, which contributed to understanding of the NRP-mediated response to rice blast. For example, soybean NRPs and their orthologs in *Arabidopsis* were found to be induced by endoplasmic reticulum (ER) stress, and triggered PCD response [12]. Ref. [29] reported *Arabidopsis thaliana NRP* genes were up-regulated in response to cold and drought stress, uncovering in their contribution in signal transduction. Thus, we inferred that *OsNRP1* may enhance rice defense through contributing to mediate ER stress.

### 2.6. Investigation of Cis-Element within OsNRP1 Promoter and Conserved Motifs in NRP Genes

*OsNRP5*, *OsNRP2*, *OsNRP8*, *OsNRP3*, *OsNRP4*, *OsNRP1,* and *OsNRP6* were found to be significantly up-regulated during interactions with rice blast fungus. To investigate the regulatory mechanisms associated with the up-regulated *OsNRP*s, we analyzed predicted *cis*-elements within their promoter regions (defined as the 1500-bp region upstream of the transcription start site) (Figure 6a). In total, 175 *cis*-elements were detected (Appendix A); of the 20 most abundant (Table 2), 8 were likely to be involved in abiotic or biotic stress responses. For example, WRKY71OS and WBOXNTERF3 are putative binding sites of WRKY proteins, suggesting that *DCD/NRP* genes may be required for WRKY-mediated regulation of plant defense against pathogens (Table 2). In addition, CGCGBOXAT, the binding site of bZIP proteins, was also present in the promoters of *NRP* genes. This result was consistent with PPI network predictions, which suggested that bZIP proteins maybe important components of the NRP-mediated cell-death signaling pathway. Moreover, the *cis*-elements OSE2ROOTNODULE and GT1GMSCAM4, which are involved in activation of pathogen- and salt-induced genes, were present; MYBCORE, ACGTATERD1, and ACGTATERD1 were found, and may participate in responses to diverse abiotic stressors, such as dehydration These findings suggested that *DCD/NRP* genes were involved not only in abiotic stress but also in plant pathogen defense, during which multiple TFs play important roles.

We also used MEME to predict conserved motifs among NPRs in *Oryza* species. In total, five conserved motifs that formed the NRP domain were found. For example, FGLP and PLxE motifs, the typical motifs of NRP proteins, were displayed as Motif_1 and Motif_5, which are present at the N- and C-terminus of NRPs, respectively, and are composed of sequences that conserve the secondary structure [17]. Thus, conserved motif organization seems to shape the NRP domain.

## 3. Discussion

The *NRPs* are a family of genes-encoding proteins that contain the NRP domain; this domain is asparagine-rich, and proteins containing it are therefore called N-rich proteins (NRPs) [17]. Ref. [16] first discovered that soybean *NRP* genes were induced when inoculated by *Pseudomonas syringae* pv. *Glycinea*, which contain the avirulence gene *avrA* [16]. However, genomic identification of *NRP* gene family members has been limited to only a few plants, such as *Arabidopsis* and soybean [29,30].

Common cultivated rice (*O. sativa*) is one of the most essential crops for food security, but its production is threatened by rice blast. Rice breeding for pathogen resistance has historically depended on a narrow range of genetic diversity, but other *Oryza* species may provide a broader range of genetic resources for breeding pathogen resistance in rice [31]. Previous reports clarified that soybean NRP proteins participate in ER stress so that soybean resistance is enhanced against several pathogens, such as *Pseudomonas syringae* pv. *Glycinea* and *Phytophthora sojae* Race 1 [16,17]. Meanwhile, the NRP domain is only present in the plant kingdom and conserved region. Therefore, we inferred that NRP proteins may also play a role in rice resistance against blast fungal, which is still unclear. To this end, we performed a genome-wide analysis of the *NRP* gene repertoires in 12 *Oryza* species; these analyses were designed to shed light on the evolutionary histories of these genes and to infer putative biological functions in rice resistance against rice blast.

As Figure 7 shows, we here conducted a hidden Markov model (HMM) search of *NRP* genes in 12 different *Oryza* species, yielding a total of 136 *NRP* genes. The number of *NRP* genes in each species ranged from nine in *O. brachyantha* to thirteen in the *O. sativa indica* group, indicating a wide distribution of *NRP* genes in *Oryza* species. Fifteen *NRP* genes were identified in soybean, which was more than were found in the *Oryza* species. This may have resulted from the previously proposed retention of extended blocks of duplicated genes in the soybean genome [32]. There was no obvious evidence of *NRP* gene family expansion in *Oryza* species. To clarify the phylogenic relationships between *Oryza NRP* genes, we constructed a phylogenetic tree from *NRP* genes in soybean and *Oryza* species. The *NRP* genes segregated into four subgroups, reminiscent of the results obtained in a previous analysis of soybean and *Arabidopsis NRP* genes [12]. No subgroup-specific gene family expansion was observed. Each subgroup contained members from all *Oryza* species included in the analysis. Taken together, these findings indicated that *NRP*s originated from the MRCA of soybean and *Oryza* species and were highly conserved during evolution.

Patterns of NRP protein domain organization were highly conserved within subgroups, but more diversified across subgroups. For example, the DCD/NRP domain of NRP proteins in subgroup III and I were preferentially found in the N- and C-, respectively. NRP proteins in subgroup II also contained Kelch motifs; these are present in the *Arabidopsis* Kelch repeat-containing F-box (KFB) protein SAGL1 and negatively regulate salicylic acid (SA) biosynthesis during immune responses [33]. Highly conserved syntenic *NRP* genes were also observed in *NRP*s in four different *Oryza* species, which was in contrast to results from the other *Oryza* species and soybean. Thus, the evolutionary trajectories of *NRP* genes appear to be influenced by species differentiation. In the *O. sativa japonica* group genome, only *OsNRP3* and *OsNRP8* showed a syntenic relationship, implying a putative paralogous relationship between *OsNRP3* and *OsNRP8*. This result also supports the finding that *NRP* gene duplication events rarely occurred in *Oryza* species. In addition, the Ka/Ks value of *OsNRP3*/*OsNRP8* suggested that these genes have undergone positive selection, which may have played a role in biological functional divergence throughout evolution.

*OsNRP1* was found that specifically up-regulated at an early stage during *O. sativa japonica* group interacting with three *M. oryzae* strains proposed by [28]. Inoculation assays of overexpression transgenic line (*OsNRP1^OX^*) revealed that *OsNRP1* slowed rice disease. A similar phenomenon was also verified in soybean. For example, *GmNRP-A* and *GmNRP-B*, the NRP proteins in soybean, was induced by the binding protein (BiP) under activation of salicylic acid (SA) signaling, and mediated programmed cell death [34].

We identified 10 candidate protein–protein interactors with OsNRP1. These interactors were found that enriched in GO terms such as ‘Response to stress’, ‘Endoplasmic reticulum unfolded protein response’, and ‘Response to organic substance’, which indicate that OsNRP1 may play a role in these biological processes. Interestingly, NRPs have been proposed to act as mediators of ER- and osmotic-stress-induced cell death in soybean [19]. ER capacity on protein procession is required for activation of the plant immune response, which relies on ER stress networks [11]. Notably, IRE1 is a critical ER stress sensor/transducer in *Arabidopsis* [35]. The TF bZIP50 was predicted as a potential interaction partner of OsNRP1. Interestingly, *OsbZIP50* is regulated by IRE1-mediated splicing and is required to regulate known marker genes involved in ER stress after pathogen recognition [36]. This result suggested that *OsNRP1* may inhibit *M. oryzae* infection through involvement in regulation of ER stress. We also established a protein–protein interaction network for AT5G42050, an *Arabidopsis* ortholog of OsNRP1; the *Arabidopsis* protein was predicted to interact with bZIP60, the *Arabidopsis* ortholog of OsbZIP50 [37]. We also found that OsNRP1 may interact with three stress-associated proteins (SAP16, SAP5, and SAP3) that have been shown to participate in multiple abiotic stress responses [38]. OsSAP1, another stress-associated protein, enhances disease resistance against virulent bacterial pathogens [39]. In addition, TF binding-site prediction revealed the presence of the CGCGBOXAT motif in the *OsNRP1* promoter. This suggested that *OsNRP1* was downstream of bZIP TFs, indirectly supporting the potential *OsNRP1*–bZIP50 interaction.

In this study, we report genome-wide analysis of NRPs across 11 *Oryza* species. Our results demonstrated conserved evolution of the *NRP* gene family, although the domain organization exhibits divergence at the subgroup level. *OsNRP1*, an *NRP* gene in the *O. sativa japonica* group, was found to be specifically up-regulated during the early stage of the rice–*M. oryzae* interaction and to inhibit *M. oryzae* infection. Moreover, predicted protein–protein interaction networks and TF binding site predictions indicated that *OsNRP1* may be downstream of bZIP TF, and may participate in the ER stress response after *M. oryzae* infection. Taken together, this study revealed *OsNRP1*-enhanced rice resistance against *M. oryzae* infection. Due to the conserved domain region and evolutionary history of NRP proteins in *Oryza* genus, this result gave an insight that *OsNRP1* can provide partial but durable rice resistance against blast fungal, which increases our understanding of biological functions of *NRP* genes in *Oryza* genus.

## 4. Method and Materials

### 4.1. Sequence Retrieval and Identification of NRP Proteins

The whole protein sequences of 12 *Oryza* species and soybean were downloaded from Ensembl Genomes (release 97; http://www.ensembl.org (accessed on 5 June 2022)) and were integrated into an initial data set for homolog identification. The hidden Markov model (HMM) file of the NRP domain was obtained from PFAM database (PF10539) [40], which was used to screen the whole proteome with cutoff E-value of 1 × 10^−5^. In parallell, known soybean NRP proteins, reported by [17], were defined as query sequences and applied BLASTP search against whole proteomes of species mentioned above (E-value cutoff = 1 × 10^−5^). Then, total putative *NRP* candidates identified by two approaches were validated by the Conserved Domains Database (CDD) database and manually removed redundant sequences.

### 4.2. Phylogenic and Evolution Analysis of NRP Family in Oryza Species

MUSCLE version 3.8.31 (Mill Valley, USA) was used to apply multiple alignment analysis of obtained NRP proteins [41]. The maximum-likelihood (ML) phylogeny trees of NRP proteins in species of this study were built with IQ-TREE version 1.6.12 (Vienna, Austria) with automatic selection of optimal model for protein substitution [42]. A bootstrap analysis with 1000 replicates was conducted to evaluate the reliability of the tree. The visualization and modification of phylogenetic trees were performed using the iTOL server [43].

### 4.3. Synteny Analysis

Genome annotations of species in this study were retrieved from Ensembl Genomes website. We used DIAMOND v0.8.25 (Tübingen, Germany) to conduct all-vs-all comparisons of corresponding protein sequences (-max-target-seqs 5 -E-value 1 × 10^−5^) [44]. Genome annotations and DIAMOND output file were input into MCScanX for synteny detection with default parameters [45]. The visualization of synteny results was performed by TBtools toolkit (Guangdong, China) [46].

### 4.4. Construction of OsNRP1^OX^ Transgenic Lines

We amplified coding sequence of *OsNRP1* (*LOC_Os01g36950*) using 2X Phata master mix (Vazyme Biotech Co., Ltd., Nanjing, China). The amplified coding sequences were cloned into the rice transformation PXQ vector. The integrated construct *PXQ::OsNRP1* was introduced into *Agrobacterium* strain EHA105 and then transformed into wild type (Zhonghua11, ZH11). Hygromycin-containing media were used to screen transgenic plants (40 mg/L).

### 4.5. Inoculation Assays

Two-week-old seedlings and leaf strips of wild-type (ZH11) and overexpression transgenic lines of *OsNRP1*, *OsNRP1^OX^* (PXQ8-4, PXQ8-5 and PXQ8-17) were inoculated by *Magnaporthe oryzae* strains Guy11. 5 mL conidia suspension were sprayed onto leaves of each seedling (5 × 10^5^ spores/mL), and 5 uL of Guy11 spore suspension was added to the wounds of each leaf strip. The disease symptoms were assessed seven days after inoculation. Leaves of ZH11 at 0, 8, 24, 48, 72 and 96 hpi were collected for real-time RT-PCR validation.

### 4.6. Real-Time RT-PCR

Qiagen RNAeasy Mini kit (Qiagen Inc., Valencia, CA, USA) was used to isolate total RNA from collected inoculation ZH11 leaves. isolated RNA samples were then converted into cDNA by the Superscript IV Reverse transcriptase cDNA synthesis kit (TB Green^®^ Premix Ex Taq^TM^ II, Takara Bio Inc, Kusatsu, Japan). We diluted a 20-μL aliquot of cDNA to 100 μL with water, which was used to established real-time RT-PCR reactions. cDNA sample of 0 hpi was used as control samples, and the actin gene of rice (*LOC_Os03g50885*) was used as an internal reference gene with a stable expression pattern as 10 housekeeping genes proposed by [47]. Bio-Rad Real-Time PCR cycler was applied for the relative gene expression level estimation using SYBG as the fluorescent dye. All primers used to real-time RT-PCR reactions were designed in Primer3 website (https://bioinfo.ut.ee/primer3-0.4.0/ (accessed on 14 June 2022)).

### 4.7. PPI Networks Prediction and Cis-Element Analysis

Protein sequence of *OsNRP1* was submitted to the online server STRING (version 11.5; http://string-db.org (accessed on 23 June 2022)), with specified organism as “*Oryza. sativa japonica* group”. Online BLAST search was used to interacting partners by detecting hits with the highest scores (Bitscore), of which Gene Ontology (GO) annotations were also provided and displayed by TBtools [46]. The *cis*-element of 8 highly-expressed *OsNRPs* in infection stage were predicted through the PLACE website (http://www.dna.affrc.go.jp/PLACE/ (accessed on 29 June 2022)), and their annotations as well. The locations of *cis*-elements were displayed by TBtools toolkit (Guangdong, China) [46].

### 4.8. Analyzing Conserved Motifs of NRP Proteins in Oryza Species

We performed multiple alignment of NRP protein sequences in *Oryza* species mentioned above by MUSCLE version 3.8.31 (Mill Valley, USA) with default parameters. Conserved motifs in NRP proteins were identified using MEME 5.0.5 according to the following parameters: -protein, -nmotifs 10, -minw 8, and -maxw 80 [48].

## Figures and Tables

**Figure 1 ijms-23-11967-f001:**
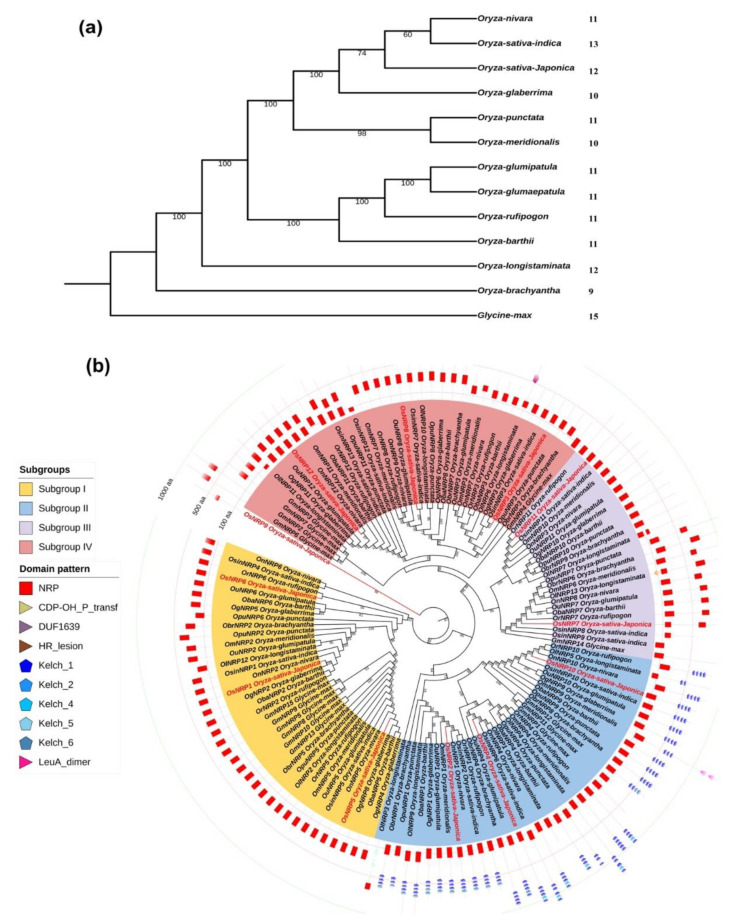
Phylogenetic relationships and distribution of *NRP* genes in 12 *Oryza* species and soybean. (**a**) Species tree of 12 *Oryza* species and soybean and distribution of NRP proteins. The species tree was constructed using the maximum likelihood method with 1000 bootstraps based on a concatenated alignment of housekeeping genes identified by CEGMA analysis. Numbers of NRP protein were listed on the right side of the species tree. (**b**) Phylogenic tree of NRP proteins identified in 12 *Oryza* species and soybean. Maximum likelihood tree, with 1000 bootstraps (values displayed per branch). NRP proteins identified in *Oryza sativa japonica* group are marked in red.

**Figure 2 ijms-23-11967-f002:**
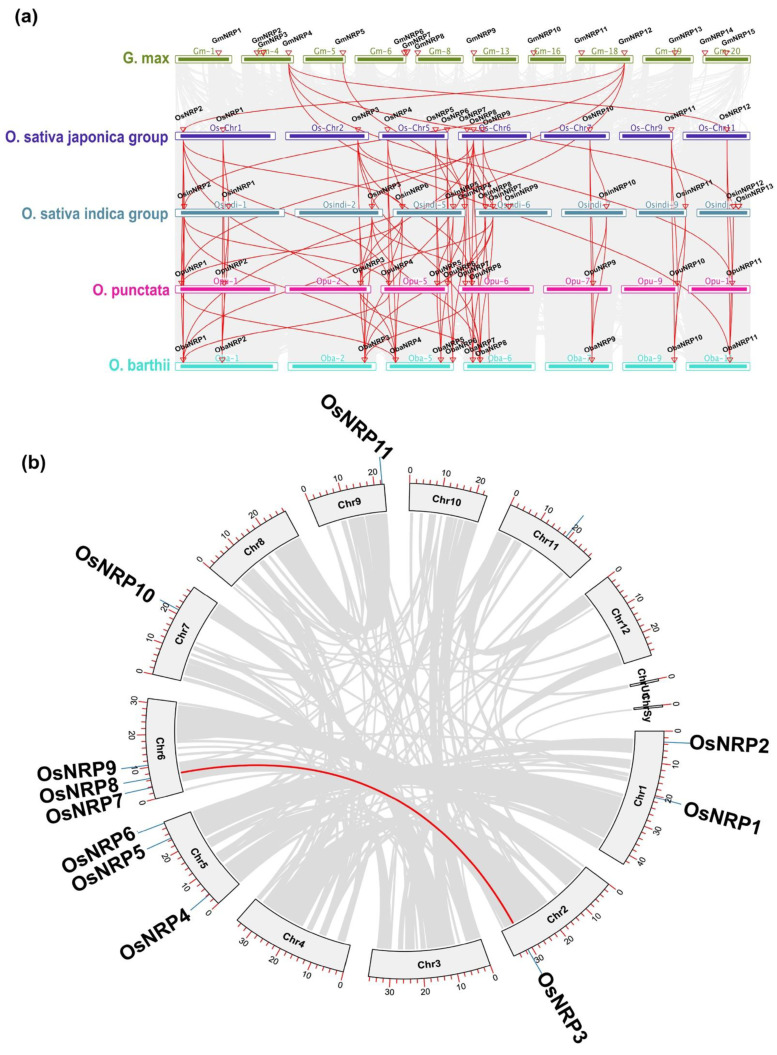
Synteny analysis of *NRP* genes. (**a**) Interspecies syntenic relationships of *NRP* genes in *Glycine max*, the *Oryza sativa japonica* group, the *Oryza sativa indica* group, *Oryza punctata* and *Oryza barthii*. Red triangles indicate the location of *NRP* genes. Red links show interspecies collinear relationships. (**b**) Intraspecies syntenic relationships of *NRP* genes in the *Oryza sativa japonica* group. Red lines show the collinear gene pairs in the *NRP* gene family.

**Figure 3 ijms-23-11967-f003:**
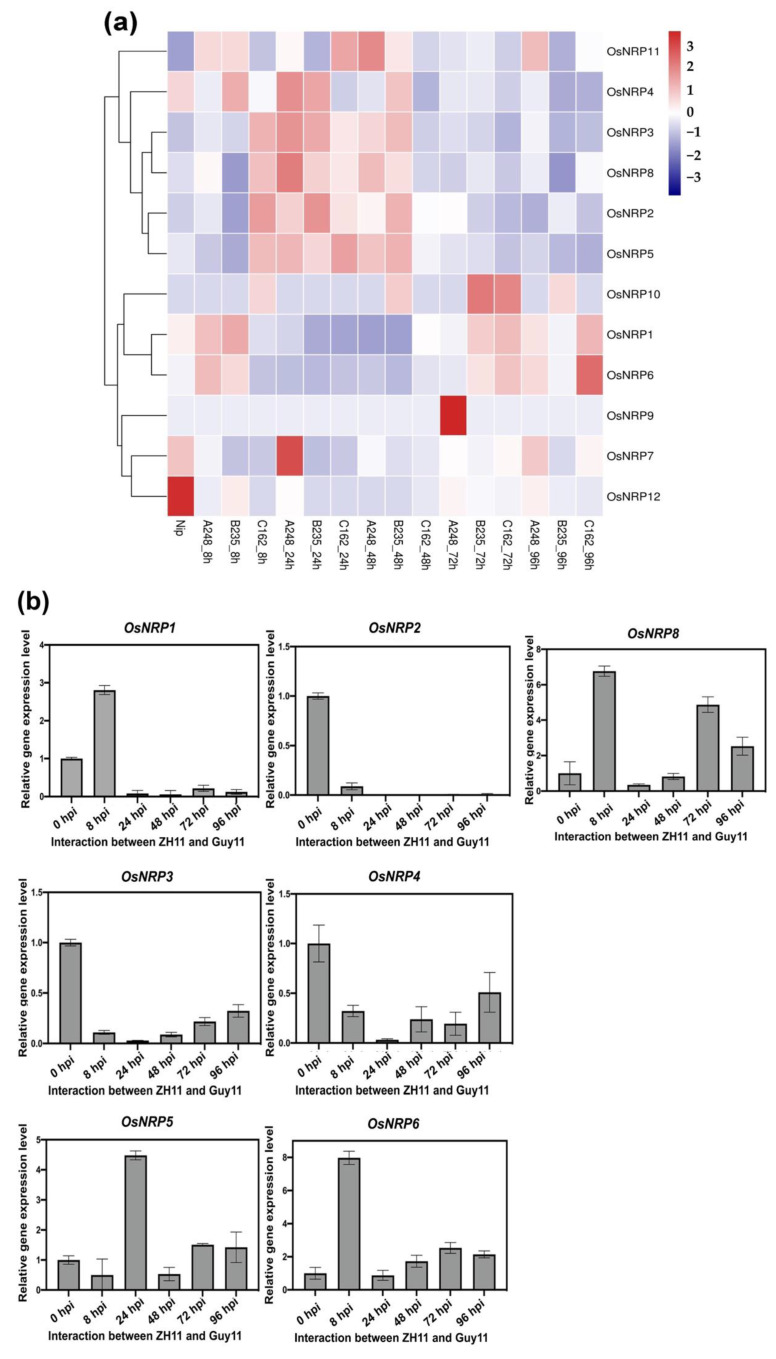
Expression analysis of *NRP* genes in the *Oryza sativa japonica* group. (**a**) RNA-seq expression profile of 12 predicted *NRP* genes in the *O. sativa japonica* group *NRP* genes during interactions with three *Magnaporthe oryzae* strains (A248, B235 and C162). (**b**) Quantitative reverse transcription (qRT-PCR) verification of selected *NRP* gene expression levels in the *O. sativa japonica* group during interactions with *M. oryzae* strain Guy11.

**Figure 4 ijms-23-11967-f004:**
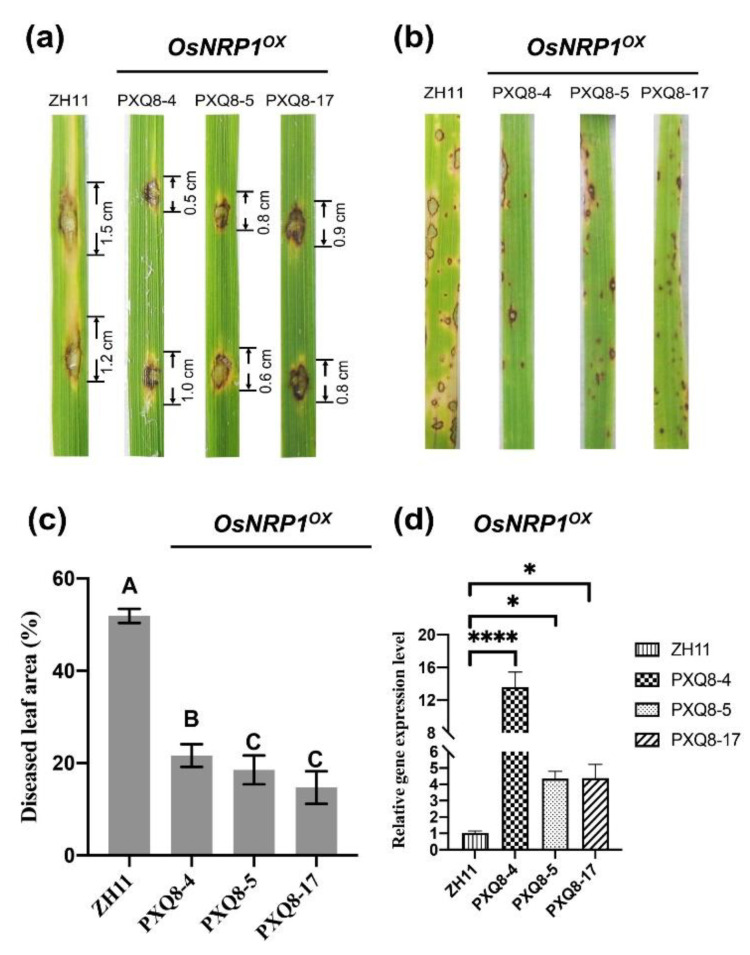
Disease reactions of ZH11 and transgenic *OsNRP1*-overexpression (*OsNRP1^OX^*) leaves incubated with *M**agnaporthe oryzae* strain Guy11. (**a**) represents in vitro inoculation. (**b**) represents in vivo inoculation. Photographs showing the disease reactions of the indicated rice lines: wild-type (ZH11) and transgenic *OsNRP1*-overexpression (*OsNRP1^OX^*) lines (PXQ8-4, PXQ8-5, and PXQ8-17). (**c**) The disease symptoms on leaves of ZH11 and *OsNRP1^OX^* to *M. oryzae* Guy11. ImageJ was used to calculate the lesion area. Different letters above bars indicate significant differences (*p* < 0.01 using one-way analysis of variance [ANOVA]). (**d**) Gene expression quantification for the gene encoding the OsNRP1 in ZH11, PXQ8-4, PXQ8-5, and PXQ8-17. * *p* < 0.05, **** *p* < 0.0001 (two-sided Student’s *t*-test, *n* = 3).

**Figure 5 ijms-23-11967-f005:**
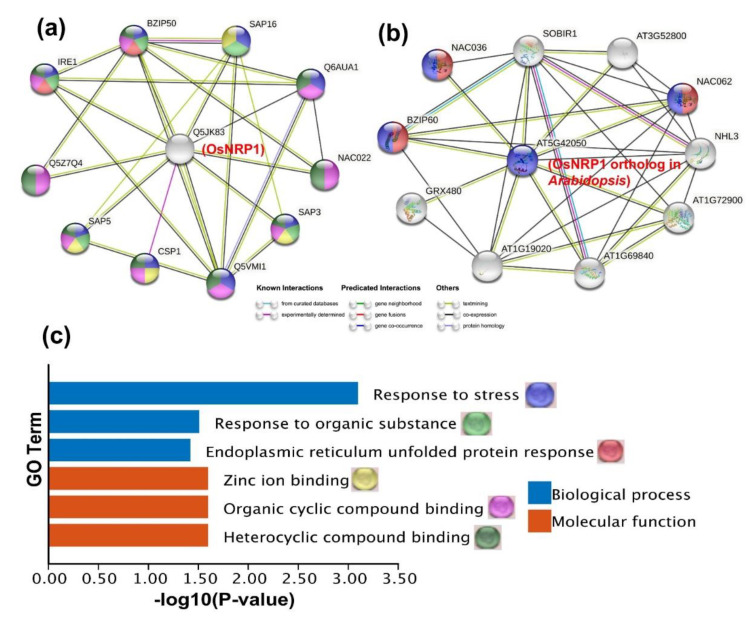
Protein–protein interaction network of OsNRP1. (**a**) Predicted protein–protein interaction network of OsNRP1 generated using STRING v11.5 website (https://string-db.org/ (accessed on 23 June 2022)) (Zurich, Switzerland). Nodes indicate OsNRP1 and its interacting candidates, and their interactions were displayed by edges with different color, which represent evidence type. (**b**) Protein–protein interaction network of the OsNRP1 ortholog in *Arabidopsis thaliana*. (**c**) Gene Ontology (GO) term enrichment analysis of proteins predicted to interact with OsNRP1. The colored circles in the right side of each term represent enriched GO terms associated with rice defense responses that are also present in (**a**) and (**b**) and represent candidate OsNRP1-interacting proteins.

**Figure 6 ijms-23-11967-f006:**
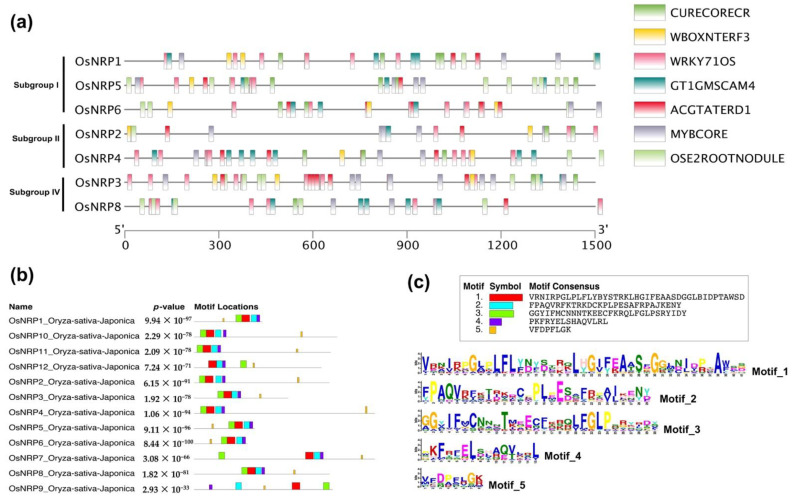
(**a**) *Cis*-element analysis in promoters of *NRP* genes in the *Oryza sativa japonica* group. The promoter of each gene was classified as the 1500-bp region upstream of the transcription start site. (**b**), (**c**) Analysis of conserved motifs in *NRP* genes in the *O. sativa japonica* group.

**Figure 7 ijms-23-11967-f007:**
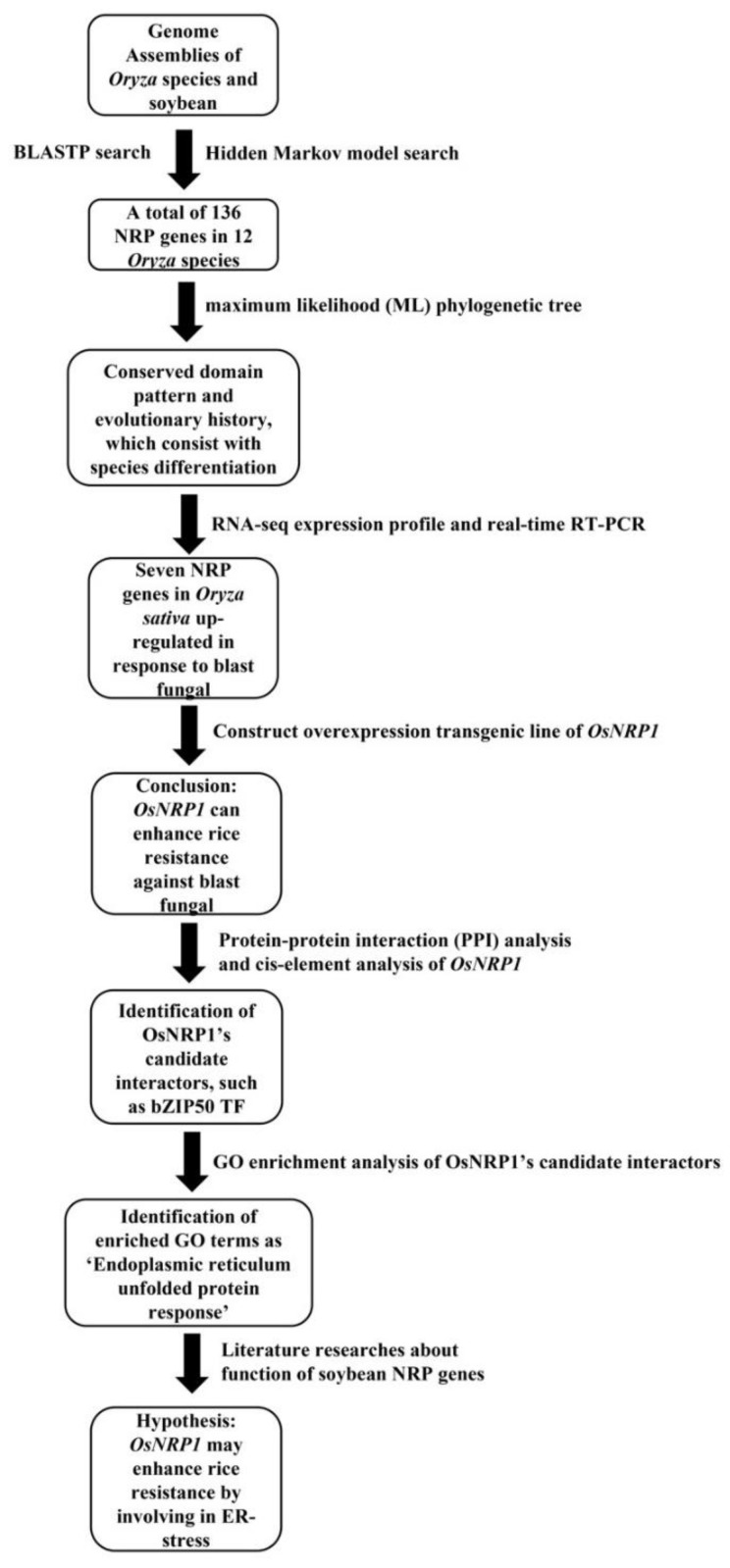
Genome-wide analysis workflow of *NRP* genes in *Oryza* species revealed *OsNRP1*’s function in rice resistance against blast fungal and hypothesis of its putative mechanism.

**Table 1 ijms-23-11967-t001:** Numbers of DCD/NRP proteins and the predicted total proteins in each genome.

Species	Chromosomes	Genome Group	NRP Proteins	Whole Proteins
*Oryza barthii*	24 (12 × 2)	AA	11	41595
*Oryza brachyantha*	24 (12 × 2)	FF	9	32038
*Oryza glaberrima*	24 (12 × 2)	AA	10	33164
*Oryza longistaminata*	24 (12 × 2)	AA	12	31686
*Oryza meridionalis*	24 (12 × 2)	AA	10	43455
*Oryza nivara*	24 (12 × 2)	AA	11	48360
*Oryza punctata*	24 (12 × 2)	BB	11	41060
*Oryza rufipogon*	24 (12 × 2)	AA	11	47441
*Oryza sativa indica group*	24 (12 × 2)	AA	13	40745
*Oryza sativa japonica group*	24 (12 × 2)	AA	12	42419
*Oryza glumipatula*	24 (12 × 2)	AA	11	46893
*Glycine max*	40 (20 × 2)	-	15	88412

**Table 2 ijms-23-11967-t002:** Top20 abundant cis-elements in promoter region of *OsNRPs*.

Motifs Accession in PLACE Database	Signal Sequence	Numbers	Functional Annotations
CACTFTPPCA1	YACT	130	cis-regulatory elements for the promoter of C4 phosphoenolpyruvate carboxylase
ARR1AT	NGATT	107	Response regulator (ARR1)-binding element
DOFCOREZM	AAAG	107	cis-regulatory elements required for binding of Dof proteins that enhance transcription of cytosolic orthophosphate kinase (CyPPDK)
GT1CONSENSUS	GRWAAW	89	cis-regulatory elements of GT-1 binding site for promoter of many light-regulated genes
CAATBOX1	CAAT	74	*cis*-regulatory elements for promoter of pea legumin gene
WRKY71OS	TGAC	65	Binding site of WRKY proteins
GTGANTG10	GTGA	61	cis-regulatory elements in the promoter of late pollen gene that is homologous to pectate lyase
POLLEN1LELAT52	AGAAA	61	regulatory elements required for specific expression of pollen gene
EBOXBNNAPA	CANNTG	56	E-box of napA storage-protein gene and R response element (RRE) responsible for light responsiveness
GATABOX	GATA	56	cis-regulatory elements in the promoter of chlorophyll a/b binding protein
MYCCONSENSUSAT	CANNTG	56	MYC recognition site in the promoters of the dehydration-responsive gene and ICE1, which involve incold stress response
ROOTMOTIFTAPOX1	ATATT	49	Motif in the rolD promoter that is highly specific to regenerating plants
CGCGBOXAT	VCGCGB	45	Motifs recognized by signal-responsive genes, like plant bZIP proteins
ACGTATERD1	ACGT	44	cis-regulatory elements involve in early response to dehydration
CURECORECR	GTAC	40	copper-response element involve in oxygen-response
WBOXNTERF3	TGACY	30	Binding site of WRKY proteins that involve in activation of ERF3 gene by wounding
MYBCORE	CNGTTR	29	MYB binding site involve in regulation of response to water-stress and flavonoid biosynthesis
NODCON2GM	CTCTT	28	nodulin consensus sequences
OSE2ROOTNODULE	CTCTT	28	cis-regulatory elements of promoters activated in infected cells of root nodules
GT1GMSCAM4	GAAAAA	27	cis-regulatory elements involve in pathogen- and salt-induced gene expression

## Data Availability

Not applicable.

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
