# Peer review of "Genome-Wide Prediction and Analysis of Oryza Species NRP Genes in Rice Blast Resistance"

_ijms, 2022, doi:10.3390/ijms231911967_

Round 1

Reviewer 1 Report

The research article by Liang et al. reported the prediction studies of the NRP gene family and provide molecular insights into rice immune responses to M. oryzae. The authors have done a commendable job performing extensive bioinformatics work and data analysis. The results and conclusions of the paper were convincing. However, I found some concerns that need to be addressed before consideration for publication.

  1. What was the rationale behind this study design? Several related studies with similar outcomes were reported previously. Unfortunately, I wasn't able to find the novelty in this report. Please discuss the importance of the study.
  2. The authors may consider adding a schematic representation of the study workflow.
  3. The “Method and Materials” section needs to be improved. The details are missing.
  4. Add a table having detailed information of the genes used in this study.
  5. The study missed several critical citations e.g. “A Genome-Wide Meta-Analysis of Rice Blast Resistance Genes and Quantitative Trait Loci Provides New Insights into Partial and Complete Resistance” by Professor Tharreau and colleagues.
  6. The figure 3-b is not clear.
  7. The authors should consider adding several ‘protein structure’ related bioinfo studies like multiple sequence alignment, molecular models, and their superimposition of NRP domains.
  8. A nit-picky suggestion- the authors could perform some pull-down assays to validate the in silico PPI study. 

Author Response

  1. What was the rationale behind this study design? Several related studies with similar outcomes were reported previously. Unfortunately, I wasn't able to find the novelty in this report. Please discuss the importance of the study.

Reply: Thanks for this good advice! 1) For the rationale behind of this study, our explanation is: Previous reports clarified that soybean NRP proteins participate in ER stress so that enhance soybean resistance against several pathogens, such as Pseudomonas syringae pv. Glycinea and Phytophthora sojae Race 1. Meanwhile, NRP domain is only present in plant kingdom and conserved region. Therefore, we inferred that NRP proteins may also play a role in rice resistance against blast fungal, which is still unclear. We also added this into newest manuscript at line 429-433, which make manuscript more logistical; 2) For the importance discuss of the study, we added discussion writing is: this study revealed OsNRP1 enhance rice resistance against M. oryzae infection. Due to the conserved domain region and evolutionary history of NRP proteins in Oryza genus, this result gave an insight that OsNRP1 have potential to provides partial but durable rice resistance against blast fungal, which increase our understanding of biological functions of NRP genes in Oryza genus. We also added this discussion writing into line 526-530 of newest manuscript.

  1. The authors may consider adding a schematic representation of the study workflow.

Reply: Thanks for this idea! We have added Figure 7 in newest manuscript to describe workflow, main conclusion and hypothesis of this study.

  1. The “Method and Materials” section needs to be improved. The details are missing.

Reply: Thanks for this advice! we have added some details missing in last manuscript. You can find it in Line 569-603 of this manuscript.

  1. Add a table having detailed information of the genes used in this study.

Reply: Thanks for this advice! we have already made a table containing detailed information of genes in this study. Because we are not sure if journal submission system displayed this Table for you, we sent this table to editor so that he or she can forward it to you. The ID of this table is Table S1.

  1. The study missed several critical citations e.g. “A Genome-Wide Meta-Analysis of Rice Blast Resistance Genes and Quantitative Trait Loci Provides New Insights into Partial and Complete Resistance” by Professor Tharreau and colleagues.

Reply: Thanks for your nice advice! We have added this citation in newest manuscript, and you can find it at line 68. And citation number here is reference [8].

  1. The figure 3-b is not clear.

Reply: Thanks for this advice! we have adjusted layout and text size of figure 3b, so that made figure 3b clear.

  1. The authors should consider adding several ‘protein structure’ related bioinfo studies like multiple sequence alignment, molecular models, and their superimposition of NRP domains.

Reply: Thanks for this advice! NRP domain is plant-specific and originally found in soybean. Until now, in silico protein structure prediction of NRP proteins mainly reported by citations of ‘DCD-a novel plant specific domain in proteins involved in development and programmed cell death’ and ‘A new cell wall located N-rich protein is strongly induced during the hypersensitive response in Glycine max L.’. Thus, we have added description of NRP protein structure at line 80-90 of newest manuscript and meanwhile cited the two publications mentioned above.

  1. A nit-picky suggestion- the authors could perform some pull-down assays to validate the in silico PPI study.

Reply: Thanks for your good advice! It will be an interesting idea to check protein-protein interactions of OsNRP1. In our future researches, we will test interactions between OsNRP1 and its putative interactors; meanwhile, the relationship of OsNRP1 and rice ER-stress and programmed cell death will also be investigated in our future researches; Moreover, we will also try to knock down OsNRP1’s interactors in our future researches (if its protein does interact with OsNRP1), which is based on OsNRP1OX to perform deep survey of OsNRP1’s molecular mechanism.

Reviewer 2 Report

Manuscript #1934823, entitled: “Genome-wide prediction and analysis of Oryza species NRP genes in rice blast resistance” presents a genome-wide analysis of NRPs across Oryza species, increasing our understanding of this gene family in their role in rice defense again Magnaporthe grisea.

More septically, it was found that the OsNRP1 gene, in the O. sativa japonica group, was specifically up-regulated during the early stage of the rice–M. oryzae interaction, inhibiting M. oryzae infection.

The research is well structured and the results are written and presented in a concise way, however, there are minor typos and grammatical errors that the authors have to take care of.

Author Response

More septically, it was found that the OsNRP1 gene, in the O. sativa japonica group, was specifically up-regulated during the early stage of the rice–M. oryzae interaction, inhibiting M. oryzae infection.The research is well structured and the results are written and presented in a concise way, however, there are minor typos and grammatical errors that the authors have to take care of.

Reply: Thanks for your kindly advices and we have correct typos and grammatical errors present in this study.

Reviewer 3 Report

In this manuscript, authors did the genome-wide prediction and analysis of Oryza species NRP genes in rice blast resistance. In this study, the authors demonstrated that the NRP genes family may have originated in the early stages of plant evolution and that they have been strongly conserved during the evolution of Oryza species. Domain organization of NRPs was found to be highly conserved within but not between subgroups. OsNRP1, an NRP gene in the Oryza sativa japonica group, was specifically up-regulated during the early rice-M stages. oryzae interactions inhibited M. oryzae infection. Predicted protein-protein interaction networks and transcription factor binding sites revealed a candidate interactor, bZIP50, which may be involved in OsNRP1-mediated rice resistance against M. oryzae infection. The overall manuscript is written well and has solid data. However, for the betterment of the manuscript, I have few suggestions for the authors.

1. Make one hypothetical figure that depicts the finding of this study.

2. The introduction is short. The author should include recent genome-wide studies such as:

a. Genome-wide identification and expression pattern analysis of the KCS gene family in barley.

b. Genome-Wide Analysis and Characterization of the Proline-Rich Extensin-like Receptor Kinases (PERKs) Gene Family Reveals Their Role in Different Developmental Stages and Stress Conditions in Wheat (Triticum aestivum L.)

c. Genome-wide identification and characterization of abiotic stress-responsive lncRNAs in Capsicum annuum.

d. Genome-wide identification and functional characterization of natural antisense transcripts in Salvia miltiorrhiza.

3. I found some plagiarized lines in the manuscript at L32-33, L36, L66-67, L68-69, L105-106, L108-109, L198, L203-205, L222-225, L273, L330, L390-391, L394-395, L396-397, L399, L404-405, L407-408, L412-414. Please clean them.

Author Response

  1. Make one hypothetical figure that depicts the finding of this study.

Reply: Thanks for this advice! we have added Figure 7 in newest manuscript to describe the analysis pipeline of this study and main finding of this study.

  1. The introduction is short. The author should include recent genome-wide studies such as:

Reply: Thanks for this advice! we have added introduction of genome-wide studies in line 104-119 of newest manuscript, which include all references you mentioned.

  1. I found some plagiarized lines in the manuscript at L32-33, L36, L66-67, L68-69, L105-106, L108-109, L198, L203-205, L222-225, L273, L330, L390-391, L394-395, L396-397, L399, L404-405, L407-408, L412-414. Please clean them.

Reply: we have corrected these lines in the newest manuscript at L31-41, L92-96, L171-176, L313-314, L317-326, L350-352, L413-414, L473-476, L569-602

Reviewer 4 Report

Liang et al conducted nice study and presented results very well.

Few comments

1. Figures need better resolution and legible text

2. It is not clear why authors selected OsNRP1, although OsNRP8 showed similar gene expression response (figure 3)

3. Need clarification of what are Figure 4A and 4B

4. Paragraphs (line 314 to 320) and (line 321 to 327) are redundant.

Thanks 

Author Response

  1. Figures need better resolution and legible text

Reply: Thanks for this advice! Yes, the Fig 1 and Fig 2 and Fig 3b is not so clear in last submission. Thus, we adjusted the layout of Fig 1 and Fig 2 based on original figures, which make more clear text in Figures and better resolution.

  1. It is not clear why authors selected OsNRP1, although OsNRP8 showed similar gene expression response (figure 3)

Reply: Thanks for this! Unfortunately, we haven’t obtained appropriate transgenic line of OsNRP8 yet. But we will continue for this in our future studies.

  1. Need clarification of what are Figure 4A and 4B

Reply: We have made separate Figure legends of Figure 4A and 4B at line 317-318 of newest manuscript.

  1. Paragraphs (line 314 to 320) and (line 321 to 327) are redundant.

Reply: Thanks for your advice! We are sorry for this mistake and removed this redundant writing. You can check it at line 467-472 of newest manuscript.

Round 2

Reviewer 1 Report

Authors have done a nice job to restructure the article. I still believe the PPI study could be part of this article. Nevertheless, the article can be considered for the acceptance. 

Reviewer 3 Report

I am happy with the revised manuscript. The authors have satisfied me with their comments. The manuscript can be accepted in its current format.